# The Impact of Cognitive Impairment on Treatment Toxicity, Treatment Completion, and Survival among Older Adults Receiving Chemotherapy: A Systematic Review

**DOI:** 10.3390/cancers14061582

**Published:** 2022-03-21

**Authors:** Schroder Sattar, Kristen Haase, Isabel Tejero, Cara Bradley, Caroline Mariano, Heather Kilgour, Ridhi Verma, Eitan Amir, Shabbir Alibhai

**Affiliations:** 1College of Nursing, University of Saskatchewan, Saskatoon, SK S7N 5E5, Canada; 2Faculty of Applied Science, School of Nursing, University of British Columbia, Vancouver, BC V6T 2B5, Canada; Kristen.haase@ubc.ca (K.H.); heatherkilgour8@gmail.com (H.K.); 3Division of Geriatric Medicine and General Internal Medicine, Department of Medicine, Sinai Health System, Toronto, ON M5G 1X5, Canada; isabel.tejero@gmail.com; 4Division of Geriatric Medicine and General Internal Medicine, Department of Medicine, University Health Network, Toronto, ON M5G 2C4, Canada; shabbir.alibhai@uhn.ca; 5Library, University of Regina, Regina, SK S4S 0A2, Canada; cara.bradley@uregina.ca; 6Department of Medical Oncology, BC Cancer Vancouver Centre, Vancouver, BC V5Z 4E6, Canada; carolinejmariano@gmail.com; 7School of Healthcare Sciences, Cardiff University, Wales CF14 4XN, UK; ridhiverma.in@gmail.com; 8Division of Medical Oncology and Hematology, University Health Network, Toronto, ON M5G 2C1, Canada; eitan.amir@uhn.ca; 9Department of Medicine, University of Toronto, Toronto, ON M5S 1A8, Canada; 10Department of Medicine, Institute of Health Policy, Management, and Evaluation, University of Toronto, Toronto, ON M5S 1A8, Canada

**Keywords:** cognitive impairment, chemotherapy, treatment toxicity, survival, treatment completion, clinical endpoints

## Abstract

**Simple Summary:**

Although cognitive impairment is common among older adults, the relationship between cognitive impairment and its effect on cancer outcomes is unclear. We conducted a systematic review of the literature to examine how toxicity risk, treatment completion, and survival may be impacted by chemotherapy in patients exhibiting evidence of cognitive impairment. Despite an absence of clear parallels between the severity of cognitive impairment and cancer outcomes, we found statistically significant relationships with survival in several studies and with toxicity in one study. An overall lack of robust evidence indicates the need for further research on the role of cognitive impairment in predicting survival, treatment completion, and toxicity among older adults receiving chemotherapy.

**Abstract:**

Cognitive impairment (CI) is common among older adults with cancer, but its effect on cancer outcomes is not known. This systematic review sought to identify research investigating clinical endpoints (toxicity risk, treatment completion, and survival) of chemotherapy treatment in those with baseline CI. A systematic search of five databases (inception to March 2021) was conducted. Eligible studies included randomized trials, prospective studies, and retrospective studies in which the sample or a subgroup were older adults (aged ≥ 65) screened positive for CI prior to receiving chemotherapy. Risk of bias assessment was performed using the Quality in Prognosis Studies (QUIPS) tool. Twenty-three articles were included. Sample sizes ranged from *n* = 31 to 703. There was heterogeneity of cancer sites, screening tools and cut-offs used to ascertain CI, and proportion of patients with CI within study samples. Severity of CI and corresponding proportion of each level within study samples were unclear in all but one study. Among studies investigating CI in a qualified multivariable model, statistically significant findings were found in 4/6 studies on survival and in 1/1 study on nonhematological toxicity. The lack of robust evidence indicates a need for further research on the role of CI in predicting survival, treatment completion, and toxicity among older adults receiving chemotherapy, and the potential implications that could shape treatment decisions.

## 1. Introduction

Cognitive impairment (CI) is defined as alterations in memory, learning, concentration, or decision making and ranges from mild to severe, with the most severe being a formal diagnosis of dementia [1]. Cognitive domains including executive function, memory, attention, and processing speed have been highlighted in the literature as among the most common impairments among individuals with cancer [2,3,4]. CI and cancer are prevalent in older adults, with over 60% of cancer diagnoses occurring in those aged ≥65 [5] and 15–48% of older adults with cancer having CI [6,7,8]. Cognitive impairment is not routinely screened by oncologists [9], and patients with CI are usually excluded from participation in clinical trials [10], which begs the question how often in everyday clinical practice are patients with CI precluded from referral to oncology or offered treatment. Older adults experiencing CI may have greater difficulty: understanding their cancer diagnosis, prognosis, and the benefits and risks associated with their treatment options; making complex decisions, including those related to cancer treatment [11]; or following instructions, all of which can potentially affect their risk of adverse events [12]. Additionally, CI, in particular dementia, can affect the potential benefits of chemotherapy (especially when given in the adjuvant setting) and also increase potential harms [11]. Even mild CI is not to be overlooked given research in the general geriatric population shows increased risk for progression to, or developing dementia, among those with mild CI [13]. To further complicate the matter, chemotherapy treatment may also lead to cognitive decline [14], and those with pre-existing CI may be particularly susceptible to further cognitive changes after receiving chemotherapy [15]. This, in turn, could portend a potential downward spiral of ability to communicate treatment-related adverse effects they experience and also have serious implications for patient safety. 

Geriatric assessment can predict treatment tolerance in older adults with cancer [16], postoperative complications [17], and mortality [18]. Importantly, CI (based on the Mini-Mental State Exam (MMSE) score) is associated with changes in cancer treatment [19]. However, the predictive power of CI, in particular, for outcomes of chemotherapy/patients treated with chemotherapy is not known. With the aging of the population and the projected increase in the proportion of older patients requiring oncology care in the foreseeable future [20], identifying ways to determine whether the patient with CI can safely receive chemotherapy and how best to safely prescribe chemotherapy has become an urgent issue to address. An important starting point is to understand how pre-existing CI impacts chemotherapy outcomes [11]. 

Although previous reviews [18,21] have examined predictors for treatment outcomes among older adults receiving chemotherapy, none specifically focused on CI. Given the increased vulnerability of older adults with cancer and the potential deleterious implications of CI in this population, older adults with concurrent CI are an even more vulnerable subset of the older cancer population. The objective of this study was to identify research investigating clinical endpoints (toxicity risk, treatment completion, and survival) of chemotherapy treatment in older patients who had CI at baseline and to examine the impact of CI on these clinical endpoints. 

## 2. Methods

### 2.1. Information Sources, Search Strategy, and Selection Process

The search strategy was developed in collaboration with and conducted by an expert research librarian (CB). Database searches included MEDLINE (including MEDLINE in process), PubMed, CINAHL, EMBASE, and PsycINFO for articles published between database inception and March 2021. See Appendix A for search strategies. All titles and abstracts were screened independently by two of five authors (S.S., I.T., C.B., K.H., C.M.). Full-text articles were then reviewed independently by two authors (S.S. and K.H.). Disagreements were resolved by discussion with a third reviewer (S.A.). If two items reporting on the same study findings were identified, the one providing more information was included. References cited in systematic reviews on similar topics were also searched. In the event that only an abstract was available, the full-text final report of the study was searched using names of the first and last authors. 

### 2.2. Eligibility

Studies eligible for inclusion:Were clinical trials, prospective cohort, retrospective cohort, or case–control studies;Included patients aged ≥65 or a subgroup analysis of patients aged ≥65;Focused on patients with a cancer diagnosis (any site, stage; with the exception of brain tumor or brain metastases) AND with CI (screened positive for CI prior to receiving cytotoxic chemotherapy or with documented medical history of CI);Examined specific chemotherapy treatment endpoints (i.e., survival, treatment completion, or treatment toxicity); andHad their full text published in English or Spanish.

Studies were deemed ineligible if outcomes of interest were not measured or reported. 

### 2.3. Data Collection Process and Data Items

Data were extracted independently by two of four authors (S.S., K.H., H.K., R.V.) and cross-checked for accuracy. All results that were compatible with each outcome domain in each study were sought (i.e., survival, treatment completion, or toxicity). Where available, we extracted the estimates (i.e., odds ratio/adjusted odds ratio or hazard ratio/adjusted hazard ratio). Data extracted also included: study design, sample size, sampling strategy, cancer type and stage, education, proportion of women in study, tool and cut-off used to assess cognition, and proportion of patients with CI within the samples. No assumption was made about any missing or unclear information from the studies. For articles requiring clarifications regarding relevant data, the corresponding authors were contacted via email.

### 2.4. Risk of Bias Assessment

A risk of bias assessment was independently performed by two of three authors on each article (S.S., K.H., and I.T.) using the Quality in Prognosis Studies (QUIPS) tool, which is commonly used to evaluate the quality of prognostic studies [21,22,23]. The QUIPS tool assesses six domains including study participation, study attrition, prognostic factor measurement, outcome measurement, study confounding, and statistical analysis and reporting. Disagreements were resolved through discussion between the two lead authors (S.S. and K.H.). The results of the assessment were presented in a summary in table format. 

### 2.5. Synthesis Methods

We summarized the results of the studies that reported on the main outcomes, as pooling findings was not feasible due to the heterogeneity among studies (i.e., in terms of patient- and cancer-related characteristics, as well as variation in assessment tools and cut-offs used), as well as the lack of studies reporting the same outcome measures. We used a tabular structure to display the results of individual studies. To synthesize study findings in each outcome of interest, we only included studies that included age and cancer disease severity/performance status as covariates in an adjusted multivariable model. To keep our reporting succinct, only key information is included in the paragraphs reporting the outcomes of interest; details of findings in each included study are reported in the Appendix A.

## 3. Results

### 3.1. Study Selections

Our systematic search resulted in 9557 titles and abstracts after removal of duplicates. See Figure 1 for the PRISMA diagram. The review was not registered.

### 3.2. Study Characteristics

A total of 17 articles remained after 69 were retrieved and subjected to full-text review, and an additional 6 reports were identified from hand searches; overall, 23 articles were included in this review [3,12,24,25,26,27,28,29,30,31,32,33,34,35,36,37,38,39,40,41,42,43,44]. The majority of the studies were conducted in the US [3,12,32,36,39], the Netherlands [25,26,27,28,35], and France [29,33,34,37,41], and were prospective cohort studies (*n* = 14). Sample sizes ranged from *n* = 31 to 703. Cancer sites explored varied, with the most common being hematological. Methods used to ascertain CI varied, and included the MMSE (*n* = 18), Informant Questionnaire on Cognitive Decline in the Elderly (IQCODE) (*n* = 3), Blessed Orientation-Memory-Concentration (BOMC) test (*n* = 2), Montreal Cognitive Assessment (MoCA) (*n* = 1), Modified Mini-Mental State (3MS) Examination (*n* = 1), Five-Word Recall (*n* = 2), and Clock-in-the-Box (*n* = 1) (some overlap exists). Cut-offs of the same tool used also varied among studies. The percentage of patients with CI within the samples ranged from 4.8% [43] to 51% [31]. 

Five studies included only patients with mild–moderate CI in their studies/analyses [12,24,31,32,40]. For the rest of the studies, it was unclear if and/or how many patients with severe CI were in the sample. 

Reporting on baseline dose reductions among study samples was uncommon. Of the studies reporting baseline dose reduction in various proportions of the samples [3,12,26,31,41,42,43], one study reported one patient receiving dose reduction due to renal impairment. For the rest of the studies, if and/or how many patients who received dose reduction/adapted dosage were among those with CI was not clear. See Table 1 for study characteristics and Table 2 for a summary of findings.

### 3.3. Risk of Bias in Studies

The risk of bias was heterogeneous but overall was low to moderate in all six domains. None of the studies were excluded based on the assessment of the risk of bias. See Appendix A for details. Of note, among studies in which CI was included in a multivariable analysis, the majority [3,12,31,36,37,39,41] adjusted for potential confounders but covariate adjustment varied. Reporting findings was heterogeneous, with the majority of studies reporting odds ratios (OR) and hazard ratios (HR), but some studies reporting only *p*-values without other estimates (i.e., OR/HR, confidence intervals). In one study [34], results from a univariable model on mortality were not reported. 

### 3.4. Survival/Mortality 

Six eligible studies [3,31,35,36,39,41] examined and reported on the association between pre-existing CI and survival (or mortality) among patients treated with chemotherapy in their MV model, with four studies reporting a statistically significant finding [3,31,36,39]. Duration of follow-up varied from 30 days to 12 months. In a retrospective case–control study by Robb et al. [39] comparing 86 patients with CI and 12 patients without CI, the non-CI group had better survival (Mdn = 72.6 months) than the CI group (Mdn = 23.0 months); *p* < 0.001 [39]. In a prospective cohort of 341 patients with hematological cancer, Hshieh et al. [3] found those with an abnormal score in the five-word delayed recall had worse median survival (10.9 (SD 12.9) vs. 12.2 (SD 14.7) months; log-rank *p* < 0.001), including when stratified by indolent cancer (log-rank *p* = 0.01) and aggressive cancer (*p* < 0.001), and in multivariable analysis when adjusted for age, comorbidities, and disease aggressiveness (OR 0.26; 95% CI 0.13–0.50). An abnormal score in five-word delayed recall was also associated with poorer survival for those undergoing intensive treatment (log-rank *p* < 0.001). On the other hand, an abnormal score for the Clock-in-the-Box test was associated with poorer survival only among patients who underwent “intensive treatment” (log-rank *p* = 0.03) [3]. Among patients with acute myeloid leukemia (AML) considered fit to receive intensive induction chemotherapy, the 30-day mortality was higher in those with CI than those without (adjusted HR 2.5 (95% CI 1.2–5.5)), median overall survival (OS) for patients with CI was 5.2 months compared to 15.6 months for those without impairment [36].

Another study on hematological patients by Dubruille et al. [31] also found impaired cognitive status was predictive of OS (HR 3.260, 95% CI 1.043–10.194; *p* = 0.042) in patients treated with chemotherapy. Others reported the 1-year OS was 63% for patients with CI versus 88% for those without [31]. A cohort study of 348 patients with various cancer types by Soubeyran et al. [41] found cognitive impairment was statistically significant in the univariable model (*p* = 0.012) for predicting survival but no longer significant in the multivariate model after adjusting for treatment site. Hamaker et al. [35] found CI to be significant among older adults with metastatic breast cancer receiving single-agent therapy in a univariate model (HR 3.74, 95% CI 1.43–9.73, *p* = 0.004); however, no multivariate modeling was conducted [35]. Of note, only one of the above studies explicitly excluded patients with severe CI [31]; for the other studies, the inclusion of patients with severe CI (and, if so, the proportion) was not clear. Of importance is that the studies reporting no statistically significant findings included at least two with relatively wide confidence intervals [25,43] with one as wide as 0.76–20.93 [25]; and several studies that did not report on estimates [24,34,35,41,44]. 

### 3.5. Chemotherapy Completion

One eligible study included patients with various cancer sites and reported an association between pre-existing CI and treatment completion [37] with higher MMSE (median 28) found to be statistically significant in a univariable analysis (*p* = 0.05) (odds ratio and confidence interval not provided) but no longer significant in the multivariable analysis [37]. One study with negative findings had a small sample size [43].

### 3.6. Chemotherapy Toxicity

One multicenter study of 518 patients reported an association between abnormal Modified Mini-Mental State Exam (3MS) score and Grade ¾ nonhematologic toxicity (OR = 0.77 [0.63–0.93]; *p* = 0.008); however, an association with hematologic toxicity was not detected [32]. Of the studies with negative findings, at least two that reported wide confidence intervals had small sample sizes [35,40]. See Appendix A for results from individual studies (including statistical findings from each corresponding study)

## 4. Discussion

This review sought to investigate how pre-existing CI influences survival, treatment completion, and toxicity among older patients receiving chemotherapy. Our results suggest CI is potentially linked to higher rates of treatment toxicity and mortality and lower rates of treatment completion in patients; however, our ability to draw definite conclusions is significantly limited by the paucity of robust data, the mixed findings, and multiple limitations among individual studies. Our findings call for further research on the topic. 

The issue of mixed findings for mortality may have in part been due to selection bias related to study designs, as patients were included and underwent assessment for CI or geriatric domains after a chemotherapy treatment plan had been made or after chemotherapy had been prescribed. Hence, the samples consisted of those who likely had already been deemed suitable for treatment by oncologists, and therefore CI may not have been a common factor that could influence the endpoints in such a context. Of note, few studies reported any upfront treatment modifications due to CI at baseline, which likely influenced outcomes. Moreover, in some of these studies, the proportion off patients in the samples with CI was as small as 7% [35]. This also may have hindered the studies’ ability to investigate the influence of CI. For instance, out of the small sample size of 55 patients in the study by Aaldriks et al. focusing on breast cancer, only 15 patients screened positive for CI (10 with the IQCODE and 5 with the MMSE, respectively) [27]. Therefore, the subsample of patients with CI may have been too small to detect any association between CI and the outcomes. 

Furthermore, although reporting a statistically significant finding, Robb et al. highlighted that their study may have been at risk of referral bias, as most of their CI patients had mild or moderate impairment as opposed to severe impairment [39]. To further complicate the issue, in the same study, 11.5% of the CI patients died within 30 days of their initial consultation compared to 1.2% in the control group. The authors of the study stated a bias might have been present because those referred to the oncology group may have been in the final stages of disease, and the level of CI may have been due to the extreme severity of cancer rather than the onset of dementia. Hence, the statistically significant finding should be interpreted in light of this context.

Because the majority of studies contained different case mixes of mild–moderate and moderate–severe CI, addressing the research question at hand is difficult, in particular given the lack of explicit delineation and analysis of subgroups based on degree of CI. The fact that no stratified analyses were done to separately examine degree of impairment (e.g., mild vs. moderate–severe CI) highlights an opportunity for future research. 

As for chemotherapy toxicity, Extermann et al. demonstrated CI’s predictive ability for nonhematologic chemotoxicity in the CRASH model [32]. However, this study excluded patients with dementia (severe CI). Hence the impact of CI may not have been fully explored. Interestingly, CI is also not included among predictors for chemotherapy toxicity in the Cancer and Aging Research Group (CARG) model [45,46]. Notably, patients in the CARG study had to provide informed consent, which likely precluded participation of those with significant CI. Few patients also screened positive for CI in their validation study. Thus, whether the dearth of statistically significant findings related to CI is due to the irrelevance of CI as a predictive factor or the lack of research and strong evidence remains unknown. Nevertheless, and more importantly, the only study that included chemotherapy toxicity excluded patients with dementia/severe CI [32]. Therefore, the impact of baseline cognition or CI, especially dementia, has not been adequately explored in research thus far. Accurately predicting risks and benefits of chemotherapy for patients with CI can be challenging without available strong data as guidance. Although findings from our review were inconclusive as to the impact of CI on chemotherapy endpoints, it does not preclude the possibility that CI may impact chemotherapy treatment endpoint outcomes. Given the known association between CI and frailty, functional decline, and mortality in older patients with cancer [47] [remains a salient factor that warrants attention and further research inquiry in the context of chemotherapy treatment. 

### Strength and Limitations

This is the first known systematic review examining the role of CI on survival, treatment completion, and treatment toxicity in older adults with cancer. Although systematic reviews have been conducted on predictors of chemotherapy intolerance [21] and the value of geriatric assessment in predicting treatment tolerability and all-cause mortality [18] our review is specifically focused on cognition and includes newer studies published since the last review. We also identified a major gap in research pertaining to the topic of CI. In addition, our review was methodologically rigorous and included a systematic search of multiple databases, titles/abstracts, and full-text screening performed using two languages, and each title/abstract and full text was independently screened by two authors. There are also important limitations. A meta-analysis could not be performed due to the heterogeneity in terms of study designs, cancer sites, and types of cognitive assessment tools and cut-off scores used. Additionally, the proportion of patients with mild–moderate vs. severe cognitive decline within each individual study was not clear. This heterogeneity also limits the generalizability of our findings from the included studies, especially given each cancer site has a distinct prognosis [29]. The lack of reporting on confidence intervals in some of the negative studies also precludes our ability to examine whether the effects were due to a true null effect, in particular in studies with small sample sizes. We did not examine delirium nor endpoints such as hospitalization and subsequent dose reductions. We also did not examine noncytotoxic systemic therapy such as immunotherapy and targeted therapy, which are increasingly being used as treatment options. 

## 5. Conclusions

Older adults are a heterogeneous population in which CI is common and complicates decisions on the best course of treatment. Given the lack of solid evidence identified in this review, more research is needed to further investigate the role of CI in predicting survival, treatment completion, and treatment toxicity among older adults with cancer treated with systemic therapy and the potential implications on treatment decisions.

## Figures and Tables

**Figure 1 cancers-14-01582-f001:**
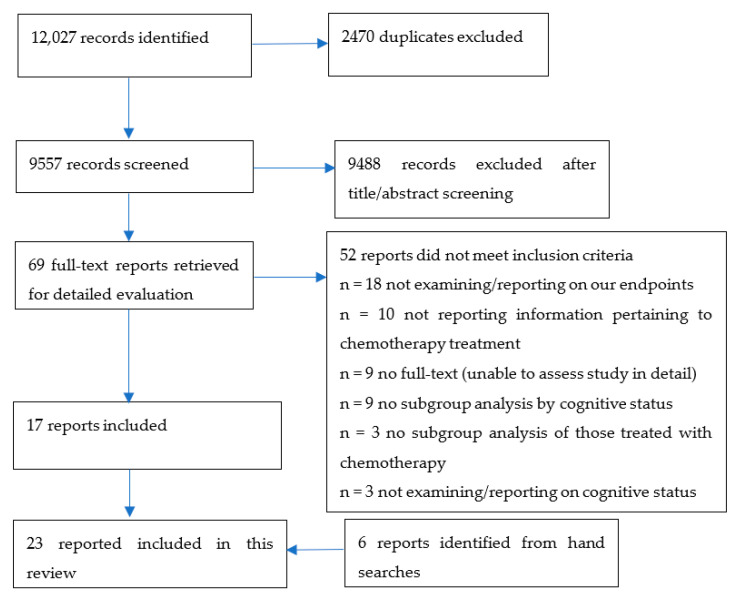
PRISMA flow diagram.

**Table 1 cancers-14-01582-t001:** Study characteristics.

Author/Year	Country	Study Type	Sample Size	Sampling Method	Cancer Site(s)	Cancer Stage/Type	Age	% Women	Educaton	Endpoint Outcomes Examined *	How CI Ascertained at Baseline
Abe (2011)	Japan	Retrospective cohort	N = 31; of whom n = 7 (22.6%) had mild/moderate CI Total 7/31 had CI (22.6%) N = 4 Mild (MMSE: 20–23) N = 3 Moderate (MMSE: 14–19)	NR	Hematological (all patients had AML)	M0-M6 (French-American-British classification)	Median 79 (total sample)	35.5% (total sample)	NR	chemotherapy discontinuation	MMSE (no cut off given) performed on patients with suspected clinically depressed cognitive function.
Aldricks (2011)	The Netherlands	Prospective	202	Consecutive	Colorectal, Hematological, breast, ovarian, upper GI, other	various	Mean 77.2 (71–92) SD 4.22	55%	NR	Chemotherapy completion Mortality	MMSE <= 24 IQCODE >= 3.3
Aaldriks (2013)	The Netherlands	Prospective cohort	143 13% positive for CI ((IQCODE >= 3.3) 8% positive for CI (MMSE <= 24)	NR	colorectal	II–IV	75 (range 70–92)	41%	NR	Mortality	MMSE ≤ 24 IQCODE > 3.31
Aaldriks (2013)	The Netherlands	Prospective cohort	55 (n = 10 [41%] positive for CI [IQCODE]) (n = 5 [13%] positive for CI [MMSE])	Consecutive	Breast	IV	76 (SD 4.8), range 70–88	96%	NR	Mortality	MMSE ≤ 24 IQCODE ≥ 3.3
Aaldriks (2016)	The Netherlands	Prospective	N = 494	Consecutive	Upper GI, Lower GI, Hematological, Breast, Gynecological, prostate, lung, urinary tract, other	I–IV	Median 75 (70–92)	50.1	NR	Chemotherapy completion (labelled as ‘feasibility’ in study) Survival	
Aparicio (2013)	France	RCT	N = 282 randomized Geriatric score for N = 123 were calculate Of whom, (n = 38 [31%] positive for CI [MMSE])	NR	Colorectal	IV	Mean 80.4 (SD 3.7)	46%	NR	Chemotherapy toxicity	MMSE ≤ 27
Biesma (2011)	The Netherlands	RCT	N = 181	NR	Lung	III–IV	Median 74 (70–87)	23%	NR	Chemotherapy toxicity	MMSE (cutoff not reported)
Dubruille (2015)	Belgium	Prospective longitudinal	N = 90	Consecutive	Hematological	NR	Median 74 (65–89)	43%	NR	One-year survival	MMSE < 27 MoCA < 26
Extermann (2012)	US	Prospective multicenter	N = 518	Consecutive	various	I–IV Lung, breast, NHL, colorectal, bladder, other	Mean 75.5 (70–92)	50.4%	NR	Hematologic Toxicity, non-hematologic toxicity	MMSE (cutoff not reported)
Falandry (2013)	France	RCT	N = 111 (29% had MMS score <25)	Consecutive	Ovarian	II–IV	Median 79 (71–93)	100%	NR	Overall survival	MMSE < 25
Falandry (2013)	France	RCT	N = 60	N/A	Breast	IV	Median 77 (71–89)	100%	NR	PFS, overall survival, chemo toxicity	5 word recall
Hamaker (2013)	The Netherlands	RCT	N = 73	N/A	Breast	IV	Median 75.5 (65.8–86.8)	100%	NR	Chemotherapy toxicity Survival	MMSE ≤ 23
Hshieh (2018)	US	Prospective observational cohort	360 341 (94.7%) completed both cognitive screening tests 127 (35.3%) had probable executive dysfunction on the CIB; 62 (17.2%) had probably impairment in working memory (5 word delayed recall)	Consecutive	Hematological	Aggressive; indolent	Mean 79.8 (SD 3.9)	35.6%	NR	Survival	Clock-in-the-Box (executive function) 7 to 8 as normal 5-word Delayed Recall (working memory) 3 of 5 words possible CI
Jayani (2019)	US	Secondary analysis of a prospective cohort study	N = 703; of whom, n = 250 (36%) had CI	Consecutive	Breast, GI, GU, gynecological, lung, other	I–IV Stage III or IV cancer (81.1%)	Mean 73 (65–94)	32.7% (out of the n = 250 group with CI)	College or higher education (63%)	Chemotherapy toxicity	Blessed Orientation-Memory-Concentration test (BOMC 5–10 as potential CI)
Klepin (2013)	US	Prospective cohort study	N = 74, 28.8% had CI	Consecutive	hematological	Cytogenic risk group: Poor: 31.5% Favorable/intermediate: 68.5%	Mean 70 (SD 6.2)	46%	< high school: 25.0% High school: 23.6% College/above: 51.4%	Overall survival	100-point Modified Mini-mental State Exam (3MS) (<77 = impairment)
Laurent (2014)	France	Prospective	N = 385	Consecutive	Colorectal, breast, upper GI + liver, urinary tract, prostate, other	Stage IV 47%	Mean 78.9 (+/−5.4)	52.2%	NR	Chemo discontinuation	MMSE < 24
Lee (2020)	Japan	Retrospective	N = 127	NR	All patients had diffuse large B-cell lymphoma	Ann Arbor Stage III/IV: 78.7%	Median 83.7 (80–96)	52.8%	NR	Survival	NR
Molga (2019)	Australia	Prospective	N = 98 (n = 11 screened positive for CI at baseline)	NR	Hematological	IPSS (international prognostic scoring system) very low to very high	77 (66–95)	36%	NR	Chemotherapy completion Overall survival	MMSE < 24
Robb (2009)	US	Retrospective case-control	CI: n = 86 Non-CI: n = 172	N/A	Breast, colorectal, prostate, gastric, pancreatic, lung, other	0-IV Stage IV 33.7%	CI: mean 79.1 (SD 5.47); non-CI: mean 75.4 (SD 4.63)	Case n= 54. Control n= 135	NR	Survival	MMSE ≤ 24
Shin (2012)	Korea	Prospective	64	NR	GI, lung, gynecological, other	I–IV Stage IV 50.0%	Median 71 (65–80)	25%	NR	Chemotherapy toxicity	MMSE-KC (Korean version) <=24 Mild cognitive decline <16 cognitive impairment
Soubeyran (2012)	France	Prospective	348	Consecutive	Non-Hodgkin’s lymphoma, GI, lung, ovarian, bladder, prostate, pancreas	Majority were stage IV (65%)	Median 77.5 (70–99.4)	40.5%	NR	Mortality (early death risk)	MMSE ≤ 23
Thibaud (2021)	Belgium	Prospective	N = 206 31% had MMSE < 27	NR	Hematological	Based on HEMA-4 score Good prognosis-Poor prognosis	Mean age 76 (65–90)	46%	NR	Survival	MMSE < 27
Wildes (2013)	US	Prospective	N = 65	Convenience	Lung, Breast, Lymphoma, colorectal	NR	Median 73 (65–89)	58.5%	NR	Chemo completion Non-hematologic toxicity Mortality	Short blessed > 9

* based on our research question (i.e., mortality, treatment toxicity, treatment completion). IQCODE = Informant Questionnaire on Cognitive Decline in the Elderly. MMSE = Mini Mental State Examination. MoCA = Montreal Cognitive Assessment. NR—not reported. N/A—not applicable.

**Table 2 cancers-14-01582-t002:** Summary of findings.

Category	Number of Studies * Investigating CI ^^^ in Multivariable Model	Number of Studies * Reporting Statistically Significant Influence of CI on Outcome
Survival/mortality	6	4 ^a^
Chemotherapy completion	1	0 ^b^
Chemotherapy toxicity	1	1 ^c^

* Studies that included age AND cancer disease severity/performance status as covariates in adjusted multivariable model; ^^^ Cognitive impairment; ^a^ An additional 2 studies found CI significant in a univariate model; ^b^ The same study only found CI significant in a univariate model; ^c^ CI was a significant predictor for nonhematological toxicity but not hematological toxicity.

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
