# Peer review of "The Impact of Cognitive Impairment on Treatment Toxicity, Treatment Completion, and Survival among Older Adults Receiving Chemotherapy: A Systematic Review"

_cancers, 2022, doi:10.3390/cancers14061582_

Round 1

Reviewer 1 Report

The authors conducted a systematic review of the literature, to examine how toxicity risk, treatment completion, and survival may be impacted by chemotherapy patients exhibiting evidence of cognitive impairment and they found statistically significant relationships with survival in several studies and toxicity in one study. 
The manuscript is interesting however there are some limitations detailed below:

Comment 1. The manuscript needs another proofreading. e.g. punctuation missing or grammatical errors.

Comment 2.  Was the review protocol published a priori?

Comment 3. The manuscript seems to follow PRISMA on some aspects but not systematically - it would be important to revamp the methods and results section to comply as much as possible with these guidelines. Please modify the flow chart following the PRISMA guidelines. Moreover in Figure 1, the authors report that they have included 17 studies but there are 23 in the text. So it is not clear how many studies are included in the systematic review.  

Comment 4. In the introduction section, the authors could better explain the definition of cognitive impairment and specify which cognitive domains are most impaired in people with a cancer diagnosis.

Comment 5. Why did the authors include articles in Spanish?

Comment 6. The authors consider only screening tests, why haven't you considered neuropsychological batteries? 

Comment 7. Do the studies included in the systematic review not consider behavioral variables?

Comment 8. The authors in the manuscript reported Table 1, but it isn't present in this paper. Moreover, the description of supplementary material in the text doesn't correspond to the table inserted in the supplementary file. Please revised and ordered the attached files.

Author Response

The authors conducted a systematic review of the literature, to examine how toxicity risk, treatment completion, and survival may be impacted by chemotherapy patients exhibiting evidence of cognitive impairment and they found statistically significant relationships with survival in several studies and toxicity in one study. 
The manuscript is interesting however there are some limitations detailed below:

We thank the review for their careful review, positive comments, and important feedback. We address the suggestions point by point below.

Comment 1. The manuscript needs another proofreading. e.g. punctuation missing or grammatical errors.

Our apologies for this oversight.  We have since proofread the manuscript and made corrections accordingly.

Comment 2.  Was the review protocol published a priori?

The review protocol was not published a priori.

Comment 3. The manuscript seems to follow PRISMA on some aspects but not systematically - it would be important to revamp the methods and results section to comply as much as possible with these guidelines. Please modify the flow chart following the PRISMA guidelines. Moreover in Figure 1, the authors report that they have included 17 studies but there are 23 in the text. So it is not clear how many studies are included in the systematic review.  

We apologize for this oversight. The full PRISMA flowchart reflecting inclusion of 23 articles has now been included. We have also updated the methods and results sections ensuring that all of the PRISMA guidelines are followed.

Comment 4. In the introduction section, the authors could better explain the definition of cognitive impairment and specify which cognitive domains are most impaired in people with a cancer diagnosis.

Thank you for this excellent suggestion. We have since added this part to the 1st paragraph in the Introduction section:

“Cognitive domains including executive function, memory, attention, and processing speed have been highlighted in the literature as among the most common impairments among individuals with cancer”

Joly F, Lange M, Dos Santos M, Vaz-Luis I, Di Meglio A. Long-term fatigue and cognitive disorders in breast cancer survivors. Cancers. 2019;11(12):1896.

Hshieh TT, Jung WF, Grande LJ, Chen J, Stone RM, Soiffer RJ, et al. Prevalence of cognitive impairment and association with survival among older patients with hematologic cancers. JAMA oncology. 2018;4(5):686–93.

Wefel JS, Kesler SR, Noll KR, Schagen SB. Clinical characteristics, pathophysiology, and management of noncentral nervous system cancer-related cognitive impairment in adults. CA: a cancer journal for clinicians. 2015;65(2):123–38.

Comment 5. Why did the authors include articles in Spanish?

In many reviews, a major limitation is that only English language articles are included. In this review, we had access to a native Spanish speaker who can review abstracts in that language, and thus we expanded our language criteria.

Comment 6. The authors consider only screening tests, why haven't you considered neuropsychological batteries? 

Cognitive screening are not typically performed on a routine basis in older adults in the oncology setting; and when screenings are performed, screening tests are commonly used.  Hence, this review did not specifically include neuropsychological batteries.  

Comment 7. Do the studies included in the systematic review not consider behavioral variables?

We are unsure what this comment was referring to. We assume the reviewer was referring to other related variables such as anxiety, depression, or fatigue. While some studies included assessment and reporting of these variables, the objective of this review was to examine cognitive impairment and, hence, we focused only on cognitive impairment.  If we have misunderstood the reviewer’s comment, we respectfully ask clarification and would be happy to address further comments.

Comment 8. The authors in the manuscript reported Table 1, but it isn't present in this paper. Moreover, the description of supplementary material in the text doesn't correspond to the table inserted in the supplementary file. Please revised and ordered the attached files.

Our apologies for this oversight.  Since the size and width of Table 1 had limited our ability to affix it directly onto the manuscript template; we attempted to upload it separately but unfortunately it was not uploaded successfully, contrary to what we had believed. Table 1 has now been included in the newly submitted zip file along with the other Supplementary materials.

In terms of the descriptions in the supplementary materials – they have now been corrected to reflect the corresponding files.  

Thank you again for your important feedback.

Reviewer 2 Report

The manuscript numbered cancers-1573258 deals with the review of scientific evidence of the role of cognitive impairment in the survival rate, completion of treatment and its toxicity among older patients receiving chemotherapy.

Although the topic is interesting, refers to the most recent issues and may be useful for both scientists and medical professionals some points and approach raise doubts. My main concern is why Authors did not prepare a meta-analysis for research selected from databases? This statistical tool allows you to determine what conclusions can be drawn from the entire publication record on a given topic, providing more accurate and broader knowledge than analyzing individual studies. That is why I strongly recommend to use this approach for the gathered publication record.

Manuscript should be subjected to major revision before considering publication in Cancers.

Some minor remarks are also presented below:

  1. Lines 85 – 87: The main aim of this review should be rewritten, as Authors after identifying research analyze them and on their basis they draw conclusions.
  2. Line 144 and Figure 1: What was the total number of articles included in this review, as there are contradictory information in the text and on the Figure. Please clarify.
  3. Line 153: ‘---‘ sign should be removed.
  4. Lines 181 – 182: What ‘five categories’ did Authors bear in mind? Please list these categories.
  5. Lines 225 – 238: Why paragraphs concerning chemotherapy completion and toxicity did not refers to all of the research included in this review?

Author Response

Comments and Suggestions for Authors

The manuscript numbered cancers-1573258 deals with the review of scientific evidence of the role of cognitive impairment in the survival rate, completion of treatment and its toxicity among older patients receiving chemotherapy.

Although the topic is interesting, refers to the most recent issues and may be useful for both scientists and medical professionals some points and approach raise doubts. My main concern is why Authors did not prepare a meta-analysis for research selected from databases? This statistical tool allows you to determine what conclusions can be drawn from the entire publication record on a given topic, providing more accurate and broader knowledge than analyzing individual studies. That is why I strongly recommend to use this approach for the gathered publication record.

We thank the reviewer for this important comment. We agree that meta-analysis is an important tool for the systematic review process especially in determining what conclusions to be drawn.  However, the heterogeneity in many aspects of the included studies precluded our ability to perform a meta-analysis. Our approach to dealing with this heterogeneity is consistent with PRISMA guidelines. This limitation of our review was highlighted in our manuscript, along with the rationale:

“We summarized the results of the studies that reported on the main outcomes, as pooling findings was not feasible due to the heterogeneity among studies (i.e., in terms of patient- and cancer-related characteristics as well as variation in assessment tools and cut-offs used), as well as the lack of studies reporting the same outcome measures.”

Manuscript should be subjected to major revision before considering publication in Cancers.

We will endeavor to undertake the required revisions to enhance the quality of this paper.

Some minor remarks are also presented below:

  1. Lines 85 – 87: The main aim of this review should be rewritten, as Authors after identifying research analyze them and on their basis they draw conclusions.

Thank you for highlighting this important part. We have since rewritten the main aim of this review to reflect our goal of identifying and analyzing research to draw conclusions.

“The objective of this study is to identify and analyze research investigating clinical endpoints (toxicity risk, treatment completion, and survival) of chemotherapy treatment in older patients who had CI at baseline, to examine the impact of CI on these clinical endpoints.” 

  1. Line 144 and Figure 1: What was the total number of articles included in this review, as there are contradictory information in the text and on the Figure. Please clarify.

Our apologies for this oversight – in particular, that the bottom section of the Figure was cutoff during the transferring process.  The full flow diagram has now been included, reflecting the inclusion of 23 articles. 

  1. Line 153: ‘---‘ sign should be removed.

The ‘---‘ sign on line 153 has been removed

  1. Lines 181 – 182: What ‘five categories’ did Authors bear in mind? Please list these categories.

Thank you for this important comment. We have since made explicit the domains included in the QUIPS assessment tool, namely, Study participation, Study attrition, Prognostic factor measurement, Outcome measurement, Study confounding, and Statistical analyzing and reporting.  We apologize for the error of mentioning five categories (domains) instead of six.

  1. Lines 225 – 238: Why paragraphs concerning chemotherapy completion and toxicity did not refers to all of the research included in this review?

We aim to report the key findings in the text to keep our reporting succinct for readers.  Hence, we reported all of the research included in this review in detail in Supplementary 3, and only highlighted the key information in the said paragraphs. We have since specified about the inclusion of complete details in the supplementary to direct readers to the table.  

Thank you again for your important feedback.

Round 2

Reviewer 2 Report

Authors sufficiently addressed the comments of the reviewer.